# *E. coli* Biomolecules Increase Glycolysis and Invasive Potential in Lung Adenocarcinoma

**DOI:** 10.3390/cancers17030380

**Published:** 2025-01-24

**Authors:** Alexis A. Vega, Parag P. Shah, Eric C. Rouchka, Brian F. Clem, Calista R. Dean, Natassja Woodrum, Preeti Tanwani, Leah J. Siskind, Levi J. Beverly

**Affiliations:** 1Department of Biochemistry and Molecular Genetics, University of Louisville, Louisville, KY 40202, USAeric.rouchka@louisville.edu (E.C.R.); brian.clem@louisville.edu (B.F.C.); 2Brown Cancer Center, School of Medicine, University of Louisville, Louisville, KY 40202, USA; parag.shah@louisville.edu (P.P.S.); calista.dean@louisville.edu (C.R.D.); natassja.woodrum@louisville.edu (N.W.); preeti.tanwani@louisville.edu (P.T.); leah.siskind@louisville.edu (L.J.S.); 3KY INBRE Bioinformatics Core, University of Louisville, Louisville, KY 40202, USA

**Keywords:** microbiome, lung cancer, glycolysis, EMT

## Abstract

Crosstalk between our cells and the microbes within our bodies has long been known to play a role in normal physiology and disruption of this crosstalk can lead to pathological conditions. We recently provided evidence that the microbes within lung cancers are enriched for methionine-producing constituents. Cancer cells can utilize this methionine under conditions of limiting nutrients. Herein, we expand these studies to demonstrate that biomolecules produced by bacteria can influence multiple signaling and metabolic pathways within lung cancer cells. By furthering our understanding of how microbes can influence tumorigenic and metastatic processes, we may be able to design novel therapeutics for treating lung cancer.

## 1. Introduction

Lung cancer is the leading cause of cancer-related mortalities in the United States [1,2]. Solid tumors, such as lung cancer, can be viewed as separate organs consisting of heterogeneous cell types that make up the tumor microenvironment (TME) [2,3,4,5,6,7,8,9,10]. While the TME was previously believed to consist solely of mammalian cells reprogrammed by the tumor, recent findings reveal that other organisms, including fungi and bacteria, also influence the TME [11,12,13,14,15].

Bacteria are single-celled organisms found throughout our bodies, and while they are known to play a protective role in our health, these commensal organisms may also become opportunistic pathogens [16] during times of dysbiosis [17]. Early studies investigating the impact of bacteria on cancer development have shown *Helicobacter pylori* as a critical player in gastric cancers [18,19]. However, *H. pylori* is only a single family that can be found in the sea of bacteria that make up our microbiome. Additionally, not only can individuals harbor their own microbiome, but the organs in our body can also encompass a different microbiome compared to other organs [20]. To add to the complexity of the microbiome, studies have shown that tumors can modulate the surrounding microbiome to potentially select for a particular microbiome that directly benefits the tumor itself [12,21]. In fact, others have shown that the removal of the microbiome in the lung using antibiotics lessens lung tumor progression and metastasis [12]. While the link between cancer and the microbiome has been established, our understanding of this relationship remains in its infancy.

Emerging research suggests that bacteria may support certain cancers, such as lung adenocarcinomas (LUADs), by providing nutrients that enhance survival and growth in competitive environments [21]. Conversely, bacteria can also act detrimentally in other cancers, such as colorectal cancer, by converting nitrogenous waste into L-arginine and enhancing PD-L1 therapy [22]. Other studies have investigated the impact of lipopolysaccharide (LPS), a bacterial toxin found ubiquitously on the cell wall of Gram-negative bacteria [23], on multiple types of cancer, showing enhanced growth and motility in breast cancer [24], increased metastasis in colorectal cancer [25], and enhanced lung tumorigenesis [26]. Given the dual roles of bacteria in cancer, it is critical to investigate their molecular interactions with tumors to better understand their impact.

In this study, we investigated the direct influence of *E. coli* biomolecules on LUAD by performing RNA-seq analysis to identify differentially expressed genes (DEGs). Our RNA-seq data revealed several phenotypic changes, including increased *JUN* expression, glycolysis, and epithelial-to-mesenchymal transition (EMT) markers. Investigating further, we saw that Hexokinase II (HKII) was significantly upregulated in the presence of *E. coli* biomolecules across multiple LUADs. To determine if this change was due to the production of LPS, we blocked the toll-like receptor (TLR) responsible for recognizing LPS, TLR4. Surprisingly, using a TLR4 inhibitor, C34, showed no reduction in glycolysis or invasive potential in LUAD cells incubated with *E. coli* biomolecules. Furthermore, we removed specific biomolecules from *E. coli*-supplemented media by treating the media with RNAse enzyme, charcoal, and dialysis. Treatment of *E. coli*-supplemented media had no affect on increased cJUN expression, while charcoal and dialysis both blocked the increase in cJUN protein levels.

Overall, our data highlight the impact of *E. coli* on LUAD cellular function including glycolysis and invasive potential.

## 2. Materials and Methods

### 2.1. Cell Culture

Human LUAD cell lines (A549, H2009, and PC9) were obtained from American Type Culture Collection (ATCC, Manassas, VA, USA). Cells were maintained in RPMI-1640 (Gibco-ThermoFisher, Waltham, MA, USA), supplemented with 10% FBS, and grown at 37 °C in a humidified 5% CO_2_ incubator. For our bacterially supplemented media, *E. coli* was grown in 2 mL lysogeny broth (LB) media for 24 h on a shaker at 37 °C. Afterwards, *E. coli* was added to RPMI-1640 supplemented with 10% FBS and 1% L-glutamine at a final OD of 0.1 and allowed to grow overnight. Following incubation, *E. coli* cells were removed from the media by centrifugation (12,000× *g* for 10 min) and the supernatant was filter-sterilized (0.2 µ); it is described herein as “bacterial supplemented RPMI-1640 media”. For inhibition of TLR4, cells were incubated in C34 (MedChemExpress, Monmouth Junction, NJ, USA) at a final concentration of 5 µM for 2 days.

Bacterially supplemented media were treated in 3 separate manners: filtered dialysis, charcoal stripping, and RNAse supplementation. A total of 14 mL of bacterially supplemented media was filter-dialyzed using Ultracel^®^-3K (Amicon, Burlington, MA, USA) at a centrifugal force of 4000× *g* for 1 h. Charcoal stripping was performed by 0.28 g of Dextran Coated Charcoal (Sigma Aldrich, Burlington, MA, USA) into 14 mL of bacterially supplemented media and incubated by rocking overnight at 4 °C. Media were then centrifuged at 2000× *g* for 15 min to remove charcoal. Bacterially supplemented media were also treated with RNAseA enzyme at a final concentration of 100 µg/mL prior to administration to LUAD cells.

### 2.2. RNA Sequencing

Human LUAD cell lines, A549 and H2009, were incubated in bacterially supplemented RPMI-1640 media for 48 h. Following incubation, RNA was isolated from LUAD cells using EZNA Total RNA Kit (Omega, Norcross, GA, USA). RNA sample integrity and concentration were determined by Agilent 2100 Bioanalyzer (Agilent Technologies, Santa Clara, CA, USA). Libraries were prepared from 1 µg RNA using the Universal Plus mRNA-Seq with NuQuant^®^ (Tecan, Männedorf, Switzerland) and Universal Plus UDI S02622 was used to barcode the libraries. Size, purity, and quantitation of libraries were performed using the Agilent Bioanalyzer (Agilent Technologies, Santa Clara, CA, USA). Libraries were quantified using the Qubit dsDNA HS kit (ThermoFisher, Q33231) and size analysis was performed with the HS NGS Fragment Kit (Agilent, DNF-474-0500) on the Agilent Fragment Analyzer. Libraries were diluted to 2 nM with Illumina Resuspension Buffer (RSB), with equivolume pooled, then further diluted to 750 pM with Resuspension buffer with Tween. PhiX positive control was spiked to a final concentration of 2%. Denaturation and final dilution were performed with an onboard instrument and libraries were sequenced using a P2 100 cycle kit (Illumina, San Diego, CA, USA 20046811) with 101 sequencing cycles and 2 index reads of 8 bp each using the NextSeq 2000 instrument (Illumina, San Diego, CA, USA). FASTQ generation was performed with BaseSpace DRAGEN analysis version 1.2.1. Raw sequencing files were downloaded from Illumina’s BaseSpace onto the KBRIN server for analysis. Sequences were directly aligned to the *Mus musculus* reference genome assembly (mm39.fa) using STAR (version 2.6) [27] to generate alignment files in bam format. Differential expression analysis was performed using two tools, Cuffdiff2 [28] and DESeq2 [29]. For the Cuffdiff2 analysis, Cuffnorm was used to produce FPKM (Fragments Per Kilobase Million)-normalized counts. The counts were then filtered to include only genes with minimum expression of one FPKM in three or more samples and an average expression of at least one FPKM. For the DESeq2 analysis, raw read counts were obtained from the STAR-aligned bam format files using HTSeq version 0.10.0 [30]. The raw counts were normalized using the Relative Log Expression (RLE) method and then filtered to exclude genes with fewer than 10 counts across the samples.

### 2.3. Measurement of Lactate Levels

For LUAD cell lines, cells were incubated in RPMI-1640 media, bacterially supplemented media, or bacterially supplemented media with C34 inhibitor for 48 h. A total of 100 µL of media was collected from LUAD plates and diluted 1:1000 with 1× PBS. Lactate measurement in diluted media was analyzed using Lactate Glo Assay (Promega, Madison, WI, USA) and luminescence was measured using SpectraMax iD3 (Molecular Devices, San Jose, CA, USA). Recorded luminescence was then normalized by cell count. For measuring lactate levels following bacterial addition, LUAD cell lines were incubated in RPMI-1640 without addition of penicillin/streptomycin for 48 h. The resulting media were collected and bacteria were added to an OD of 0.1. Bacteria were incubated on a rocker for 24 h at 36 °C. Media were then collected, centrifuged at 12,000× *g* for 10 min, and filter-sterilized with 0.2 µ filter. The resulting media were diluted 1:1000 with 1× PBS and measured using the Lactate Glo Assay.

### 2.4. Measurement of Glucose Levels

LUAD cell lines were incubated in RPMI-1640 media. The media were then collected, centrifuged (300× *g* for 4 min), and filter-sterilized using a 4 µm pore-sized filter. The resulting media and fresh RPMI media were diluted 1:100 followed by measurement using Glucose Glo Assay (Promega, Madison, WI, USA). Luminescence was measured using SpectraMax iD3 (Molecular Devices, San Jose, CA, USA). The resulting luminescent was then compared against a standard curve ranging from 50 µM to 0.01 µM glucose to determine the final glucose concentration.

### 2.5. Deoxyglucose Measurement

LUAD cells were incubated in a 6-well dish with RPMI-1640 or bacterially supplemented media. [^3^H]-labeled deoxyglucose (PerkinElmer, Waltham, MA, USA) was added to LUAD cell lines at 2 μCi/mL following incubation for one hour. LUAD cells were harvested, washed with 1× PBS, and lysed with 1% SDS in RPMI-1640. Radioactivity of the cell lysate was measured by scintillation counter, with counts per minute being normalized based on BCA protein concentration.

### 2.6. Wound-Healing Assay

LUAD cell lines were seeded in a 6-well dish at a confluency of 2 × 10^6^ per well. The next day, a wound was created using a pipette tip and layered with 200 µL Matrigel (Corning, Glendale, AZ, USA) at a final concentration of 8 mg/mL infused either with RPMI-1640 or bacterially supplemented media. LUAD cells were then incubated in RPMI-1640 media for 72 h and cellular motility was captured every 24 h. Wound closure was determined by ImageJ (version 1.53T), normalizing the percentage wound closure by wound area at 0 h.

### 2.7. Migration and Invasion Assay

Transwell inserts with an 8 µm pore size (Corning, Glendale, AZ, USA) were used for measuring chemotaxis and LUAD invasive potential. For investigating bacterial biomolecules as a chemo-attractant, 1 × 10^5^ LUAD cells were seeded on the insert. The top chamber contained RPMI-1640 media, while the bottom chamber contained either RPMI-1640 media or bacterially supplemented media. Cells were incubated for 24 h, followed by fixing with ice-cold 100% methanol and removal of unmigrated cells. Cells were then stained using Hema 3 (Fisher Scientific, Hampton, NH, USA), captured using a BZX-800 microscope (Keyence, Osaka, Japan), and analyzed using a BZX-800 analyzer.

For investigating invasive potential, inserts were hydrated with ice-cold 1× PBS; then, 200 µL of Matrigel at a final concentration of 8 mg/mL was layered atop the membrane and incubated at 37 °C incubator for 1 h. A total of 5 × 10^4 LUAD cells^ were then seeded on the top chamber and placed in either serum-starved RPMI-1640 media, serum-starved bacterially supplemented media, or serum-starved bacterially supplemented media with C34 inhibitor. The bottom chamber contained RPMI-1640 supplemented with 10% FBS. Cells were incubated for 48 h and then fixed with ice-cold 100% methanol, followed by the removal of non-invasive cells. Cells were then stained using Hema 3, captured using the BZX-800 Keyence microscope, and analyzed using the BZX-800 analysis software (version 1.1.30).

### 2.8. Immunoblotting

Whole-cell lysis was performed using 1x CHAPS buffer containing protease and phosphatase inhibitors. A total of 30 µg of lysates ran on a NuPAGE 4 to 12% Bis-Tris PAGE gel and was transferred to a PVDF membrane. Membranes were blocked overnight with 5% milk in TBS-T and probed with anti-ZO1 (1:5000), E-cadherin (1:5000), Claudin-1 (1:5000), SNAI1 (1:5000), cJun (1:5000), Hexokinase I (1:5000), Hexokinase II (1:5000), Lactate Dehydrogenase A (1:5000), and Pyruvate Kinase M2 isoform (1:5000) antibodies overnight at 4 °C. Membranes were then washed and incubated with HRP-conjugated anti-rabbit (1:10,000) secondary antibody for one hour. All antibodies were procured from Cell Signaling Technology at Danvers, MA, USA. ECL chemiluminescent reagent (Thermo Fisher) was used to detect protein levels. Membranes were also probed with anti-GAPDH (Cell Signaling Technology, Danvers, MA, USA) as a loading control.

### 2.9. Real-Time qPCR

Total RNA was isolated from cell pellets using E.Z.N.A Total RNA Kit I (OMEGA Bioteck Inc., Norcross, GA, USA) according to the manufacturer’s protocol. A total of 1 µg of isolated RNA was converted to cDNA using a High-Capacity RNA-to-cDNA kit (Applied Biosystems, Waltham, MA, USA). Gene expression was determined by qPCR using SYBR green and the following primers: *cJun* Forward: AACGACCTTCTATGACGATGCCCTC; *cJun* Reverse: GCGAACCCCTCCTGCTCATCTCTC; *HK2* Forward: GATTTCACCAAGCGTGGACT; *HK2* Reverse: CCACACCCACTGTCACTTTG; *AKR1B10* Forward: CCCAAAGATGATAAAGGTAATGCC; *AKR1B10* Reverse: TCAGTCCAGGTTTGTTCAAGAGC; GAPDH Forward: GTCTCCTCTGACTTCAACAGCG; GAPDH Reverse: ACCACCCTGTTGCTGTAGCCAA; β-*Actin* Forward: CTCTTAATGTCAGGCACGAT; β-*Actin* Reverse: CATGTACGTTGCTATCCAGGC.

### 2.10. Combination KEGG Analysis

DEGs observed in A549 and H2009 cells were combined, sorted, and ranked using Python (version 3.9.13), with pandas (version 1.4.2), matplotlib (version 3.5.1), and bioservices (version 1.11.2). Script can be provided upon request.

### 2.11. Statistical Analysis

All statistical analysis was performed using GraphPad Prism (version 10.1.2). Data are reported as average ± SEM, wherein *p*-values, statistical tests, and replicates are described in respective figures.

## 3. Results

### 3.1. E. coli Biomolecules Elicit a Number of Transcriptomic Changes in LUAD Cell Lines

Our previous work has shown that *E. coli* can secrete biomolecules that LUAD cells can utilize [21]. To gain a better understanding of the impact of *E. coli* biomolecules on LUAD cells, we treated A549 and H2009 cell lines with media containing *E. coli* biomolecules and performed RNA sequencing (Figure 1A). Our results show over 1000 differentially expressed genes (DEGs) in A549 cells and over 5000 DEGs in H2009 cells. Of these DEGs observed, 724 DEGs were upregulated and 347 DEGs were downregulated in A549, while 2715 DEGs were upregulated and 2543 DEGs were downregulated in H2009 cells. We isolated DEGs that were upregulated and combined both datasets to show that 948 genes were upregulated in both A549 and H2009 (Figure 1B). Of the 948, we observed several genes related to glycolysis/gluconeogenesis and the Hippo Pathway (Figure 1B). We went on to investigate the pathways associated with these common DEGs and ranked the KEGG pathways based on the number of genes related to that pathway (Figure 1C). Not only did we observe metabolic pathways as the most upregulated KEGG pathway observed in the presence of *E. coli* biomolecules, but we also witnessed glycolysis/gluconeogenesis being within the top 40. We do not consider that the influence of *E. coli* bacterial factors on some of these pathways is an artifact of the selection we performed, as some of these pathways are also among the top 20 most upregulated KEGG pathways of individual LUAD cell lines and the PCA analysis showed clear separation between cell and treatment groups (Figure 1D,E).

Investigating upregulated genes related to glycolysis/gluconeogenesis further showed that addition of *E. coli* biomolecules led to increases in Hexokinase I (*HKI*), Hexokinase II (*HKII*), Lactate Dehydrogenase (*LDHA*), Pyruvate Kinase M isoform (*PKM*), 6-phosphofructo-2-kinase/fructose-2,6-biphosphatase 3 and 4 (*PFKFB3/4*), Aldolase (*ALDOA* and *ALDOC*), Phosphoglycerate kinase 1 (*PGK1*), Phosphoglycerate mutase 1 (PGAM1), and Enolase 2 (*ENO2*) (Figure 1B). Furthermore, we observed *PFKFB3/4* within the top 10 upregulated DEGs in A549 cells and within the top 20 upregulated DEGs in H2009 (Figure 1C). Both *HKII* and *PKM* are irreversible steps that are only found in glycolysis and not in gluconeogenesis. Meanwhile, we were unable to identify enzymes that were considered irreversible in gluconeogenesis. As such, we believe that *E. coli* has a direct impact on LUAD glycolysis.

The Hippo Pathway is a conserved signaling pathway that is involved in several cellular processes such as proliferation, metabolism, and cancer development [31]. Previous studies have also linked the Hippo Pathway with EMT in cancer [32]. The shared genes between cell lines involved in the Hippo Pathway include FYVE, RhoGEF, PH Domain-Containing 4 (*FGD4*), LDHA, Snail Family Transcriptional Repressor 1, 2, and 3 (*SNAI1/2/3*), Claudin Domain-Containing 1 (*CLDND1*), Cyclin E1 (*CCNE1*), Jun Proto-Oncogene, AP1 Transcription Factor Subunit (*JUN*), and Ras Association Domain Family Member 5 (*RASSF5*). Additionally, we see downregulation of EMT regulators such as Aldo-Keto Reductase Family 1 Member B10 (*AKR1B10*) [33], Salt inducible kinase 1B (*SIK1*) [34], and Tight Junction Protein 3 (*TJP3*) [35]. With this in mind, we chose to investigate the effects of *E. coli* biomolecules on EMT.

### 3.2. E. coli Biomolecules Increase Glycolysis in LUAD Cell Lines

It is common knowledge that cancer cells upregulate glycolysis and shuttle pyruvate into lactate, even in oxygen-rich environments. Gene set enrichment analysis (GSEA) of our RNA-seq data revealed positive enrichment of glycolysis in LUAD cell lines incubated with *E. coli* biomolecules (Figure 2A). Since our RNA-seq data indicated elevated transcriptional levels of several glycolytic enzymes, we next examined whether these genes were also upregulated at the translational level. LPS, a major component of the Gram-negative bacterial cell wall, has previously been shown to increase glycolysis in colorectal cancer [36]. To assess whether LPS was driving the increased glycolytic expression in our LUAD cell lines, we included a treatment group of cells with media containing 0.5 µg/mL of LPS. Our results showed that *E. coli*-supplemented media, but not LPS addition, dramatically increased HKII expression, but not that of HKI, PKM2, or LDHA (Figure 2B,C). Furthermore, as *E. coli* biomolecules resulted in both the transcriptional and translational regulation of HKII, we investigated the rate of glucose uptake in LUAD cell lines using deoxyglucose, an analog of glucose unable to be metabolized past HKII that can be detected by radiometric means. After normalizing our scintillation counts using protein concentration, we saw an increase in glucose uptake in LUAD cell lines following incubation with *E. coli* biomolecules (Figure 2D).

As mentioned, cancer cells are known to favor lactate production over directing pyruvate into the citric acid cycle. Additionally, glycolytic intermediates can be diverted into other pathways, such as the pentose phosphate pathway or the serine synthesis pathway, to support the synthesis of the biomolecules necessary for cellular growth. Given the increase in glucose uptake observed in our data, we performed a luciferin-coupled assay to measure lactate concentrations in the media. This was performed to determine whether glucose was being fully metabolized through glycolysis or redirected into alternative pathways. Interestingly, we observed a dramatic increase in lactate production in all three LUAD cell lines incubated with *E. coli*-supplemented media, while no significant increase was detected with LPS treatment compared to the vehicle control (Figure 2E). These findings suggest that *E. coli* enhances glucose uptake and glycolysis in LUAD cells, leading to increased lactate production. This raised the following question: Is the increased glycolysis simply an innocuous byproduct of *E. coli* activity, or is *E. coli* benefiting from the elevated lactate levels?

Previous reports have suggested bacteria capable of utilizing the nitrogenous waste byproduct of cancer cells [22]. Additionally, others have shown a direct link between cancer and the bacteria [14,21]. We sought to determine whether lactate produced by LUAD cell lines was also being taken up by *E. coli*. We measured the lactate levels found in media previously used to grow LUAD cell lines, before and after 24 h incubation with bacteria, using the luciferin-coupled lactate assay and witnessed a dramatic decrease in lactate levels (Figure 3A).

To determine whether the increase in lactate uptake from the bacteria was due to LUAD depletion of glucose, we first investigated the concentration of glucose remaining in RPMI media incubated with LUAD cell lines. We calculated 3.8 mM glucose remaining in the RPMI media from our glucose-glo assay (Figure 3B). With this information, we added glucose to a final concentration resembling that of our RPMI media, 10.5 mM glucose. We then compared the lactate uptake by *E. coli* incubated in RPMI media previously incubated for 24 h with LUAD cell lines, with and without additional glucose supplemented, after the removal of the LUAD cell lines. From our data, we see that the *E. coli* cells take up similar levels of lactate regardless of the levels of glucose in the media (Figure 3C). Given that several reversible steps in glycolysis were upregulated in our RNA-seq data, we investigated whether *E. coli* was recycling lactate back into glucose. Interestingly, we observed a modest but significant increase in glucose levels in the media following *E. coli* incubation (Figure 3D). To explore whether *E. coli* was producing another key nutrient for cancer progression, glutamine, we measured glutamine levels in the media. While LUAD cells depleted glutamine, we did not observe a replenishment of glutamine after bacterial incubation (Figure 3E). Notably, the uptake of lactate by *E. coli* did not lead to increased survival or proliferation of the bacteria. These findings suggest that *E. coli* may reprogram cancer cells to enhance glycolysis, resulting in increased lactate production, which is then released into the extracellular matrix and taken up by the bacteria. However, any potential benefits that *E. coli* might derive from this interaction remain unclear.

### 3.3. Bacterial Biomolecules Enhance Invasive Potential of LUAD Cell Lines

The previous literature has shown that bacteria can play a role in cancer metastasis, such that antibiotic treatment diminishes lung cancer metastasis, while inoculating germ-free mice with bacteria increased lung cancer metastasis [12]. As our RNA-seq KEGG data showed the Hippo Pathway to be upregulated in LUAD cell lines incubated in bacterially supplemented media, and the Hippo Pathway is related to EMT signaling, we chose to investigate the impact of bacterial biomolecules on LUAD metastatic potential. We first analyzed our RNA-seq data against GSEA and observed enrichment of genes related to EMT in LUAD cell lines incubated in bacterially supplemented media (Figure 4A). We continued this investigation by analyzing whole-cell lysates through Western blot and probing for EMT specific markers. From our DEG results, we identified *SNAI1,2,3* and *CLDND1* to be transcriptionally upregulated in the presence of *E. coli* biomolecules. Our Western blot analysis further validated these findings, showing an increase in Claudin and SNAIL (Figure 4B). Furthermore, we saw a decrease in expression for E-cadherin and ZO1, known epithelial markers.

As cancer cells transition into a mesenchymal state, they acquire several properties, including increased motility. To assess this, we examined the expression of Claudin and SNAIL and investigated whether these changes resulted in increased motility. We performed a modulated wound-healing assay, in which the wound was coated with Matrigel diluted in either RPMI-1640 media (vehicle) or *E. coli*-supplemented biomolecules (bacteria). Our results revealed no difference in wound-healing potential between the vehicle and bacteria-treated groups (Figure 4C,D). Additionally, using a Keyence live-cell imager, we analyzed single-cell motility and observed no significant change in the distance traveled by LUAD cells incubated with *E. coli* biomolecules (Figure 4E). These findings align with previous studies, which have demonstrated that eliminating the tumor microbiome, rather than the gut microbiome, reduces metastatic nodes in breast cancer mouse models [37].

Although we did not observe changes in motility in the wound-healing assay or live-cell imaging, we further investigated the invasive potential of LUAD cells in the presence of *E. coli* biomolecules. Using a transwell invasion assay, we employed FBS as a chemo-attractant and serum-starved LUAD cells in RPMI-1640 media, with or without *E. coli* biomolecules. After 48 h, we observed a significant increase in the number of invading LUAD cells in the presence of *E. coli* biomolecules (Figure 4F,G). Collectively, our data suggest that while *E. coli* biomolecules do not enhance motility, they significantly promote the invasive potential of LUAD cells by stimulating epithelial-to-mesenchymal transition (EMT).

### 3.4. Bacterially Induced LUAD Phenotype Observed Is TLR4-Independent

Our data provide clear evidence that bacterial biomolecules directly impact the phenotype of LUAD cell lines. However, the mechanism underlying the observed increase in glycolysis and induction of EMT remained unclear. Previous studies have implicated lipopolysaccharide (LPS)-induced Toll-Like Receptor 4 (TLR4) activation in promoting proliferation and metastasis in various cancers [38,39,40,41,42]. TLR4 activation is known to enhance the MAPK signaling pathway, leading to upregulated HIF1α expression and activity, which in turn increases HKII expression [43,44,45,46]. Additionally, MAPK activation has been linked to increased expression of the proto-oncogene *JUN* [47]. Our RNA-seq results showed that *JUN* expression was upregulated in both cell lines treated with bacterially supplemented media. Therefore, we hypothesized that the effects of *E. coli* on LUAD cell lines might be mediated through LPS and TLR4 activation and tested this by blocking TLR4.

C34 is an amino monosaccharide that has been shown to inhibit TLR4 [48,49]. Previous investigators have reported a decrease in PMA-induced colorectal cancer proliferation following treatment with 5 µM C34 [50]. Using alamarBlue, we assessed the viability of A549 or PC9 following treatment with increasing concentrations of C34 and observed no toxicity when using the previously reported concentrations (Figure 5A). However, H2009 cells were sensitive to C34 treatment, showing decreasing cellular viability at concentrations of 2.5 µM. Given that TLR4 activation is associated with MAPK pathway phosphorylation, we treated LUAD cell lines with varying concentrations of C34 to identify the appropriate dose for effective TLR4 inhibition. At a concentration of 5 µM, we observed a marked reduction in p-MAPK levels (Figure 5B). Based on these results and the observed viability changes in H2009 cells, subsequent experiments were conducted using 5 µM C34. To assess the impact of TLR4 inhibition on HKII expression, we treated LUAD cell lines with C34. No significant changes in HKII expression were detected (Figure 5C). Furthermore, lactate levels in the media of cells treated with C34 did not show a significant decrease, indicating that TLR4 activation does not appear to drive the enhanced glycolysis observed in the presence of *E. coli* biomolecules (Figure 5D). These findings suggest that while TLR4 signaling may play a role in some cancer-related processes, the increased glycolysis observed in LUAD cell lines treated with *E. coli* biomolecules occurs independently of TLR4 activation.

We next broadened our approach by investigating the type of biomolecules secreted by *E. coli* that were eliciting the changes observed by removing key biomolecules. RNA was removed from our *E. coli*-supplemented media using RNAseA (R), lipids were removed using charcoal (C), and molecules >3 kDa in size were removed by dialysis (D). Our Western blot data show that charcoal stripping or dialyzing our *E. coli*-supplemented media led to decreased levels of HKII, Claudin, and cJun (Figure 6A). We performed qPCR to investigate the expression levels of *JUN*, *HKII*, and *AKR1B10*, a gene that our RNA-seq data show to be downregulated by the presence of *E. coli* biomolecules. Interestingly, we observed that either charcoal stripping or removal of biomolecules >3 kDa resulted in downregulation of *JUN* and *HKII* and upregulation of *AKR1B10* (Figure 6B). Overall, we believe that *E. coli* cells are stimulating LUAD through small lipid-based molecules, enhancing glycolysis and facilitating EMT in LUAD, wherein the lactate production is taken up by the *E. coli* (Figure 6C).

## 4. Discussion

Recent studies have linked the tumor microbiome with cancer progression and metastasis [12,13,14,18,22,51]. These studies have not only highlighted the importance of the tumor microbiome on cancer but have also suggested the possibility of direct crosstalk between the tumor cells and the microbiome. In this study, we further evaluated the impact on the microbiome directly onto the LUAD by using RNA-seq to identify transcriptomic changes that may highlight pathways that are tied with bacteria. Of the differentially genes expressed in LUAD cell lines exposed to *E. coli* biomolecules, several of these genes have been related to glycolysis and the Hippo Pathway.

It is well known that cancer cells are highly addicted to glucose and shuttle pyruvate into lactate even in aerobic environments, a phenomenon known as the Warburg Effect. While not much is known about why the Warburg Effect occurs, many have speculated that the reason behind this is shuttling glycolytic intermediates to different pathways. With the results from our RNA-seq data, we investigated the impact of *E. coli*-secreted biomolecules on glycolysis in LUAD. Not only did we witness increased elevation of HKII protein, but we also observed an elevation in both glucose uptake and lactate production. These data would suggest that the presence of bacteria facilitates upregulation of glycolysis in LUAD. Previous findings have investigated the role of nitrogenous waste produced by tumors on bacteria and found that *E. coli* utilized the nitrogenous waste to generate arginine for immune cells. Previously, it was believed that lactate was another waste product of cancer cells but has been shown to play a role in signaling and metastasis and is even a fuel source for cancer cells [52]. With this in mind, we sought to investigate whether the lactate produced by cancer cells could also be taken up by *E. coli*. We observed a dramatic decrease in lactate in our LUAD media after incubating with *E. coli,* suggesting that the bacteria were indeed taking up the lactate.

Beyond glycolysis, our RNA-seq data suggested that *E. coli* biomolecules influence the Hippo Pathway, which governs phenotypes such as survival and motility. By analyzing the top 20 downregulated DEGs, we identified several genes linked to cellular adherence, prompting us to focus on LUAD motility and invasion. Protein analysis showed increased SNAI1, slight upregulation of Claudin, and reduced E-cadherin expression, indicating epithelial-to-mesenchymal transition (EMT). Using a modified wound-healing assay with Matrigel infused with *E. coli*-supplemented media, we found no evidence of bacterial biomolecules acting as chemo-attractants. However, invasion assays revealed a significant increase in LUAD invasive potential toward fetal bovine serum when cells were serum-starved and exposed to *E. coli* biomolecules. These findings align with prior studies showing that eliminating the microbiome around tumors diminishes metastatic potential in cancers like breast cancer [37].

As our study explored the direct relationship between LUAD and bacteria, we next wanted to determine how *E. coli* was eliciting the phenotypic changes in LUAD cell lines. As *E. coli* are Gram-negative bacterial cells, their cell wall has a large abundance of LPS. LPS is known to impact immune cells by activating the TLR receptors, namely TLR4, on the surface of immune cells and leading to phenotypic changes such as increased glycolysis. Cancer cells are also known to have TLR4 receptors where activation has been shown to increase survival, proliferation, and even metastasis [40,41,42,53,54]. To determine whether the effects we were observing were related to stimulation of the TLR4 receptor, we used a potent TLR4 inhibitor known as C34. As C34 had not previously been investigated in cancers, we first determined the potency of the inhibitor on LUAD cell viability. We saw no dramatic decrease in cell viability for two of the three cell lines chosen. Of the decrease in viability observed in our H2009 cell line, we saw the decrease in viability begin at 5 µM and level off at around 50% viability. We continued with our experiments, using 5 µM C34 to determine the effects of TLR4 inhibition on glycolysis and invasion. We observed no change in expression levels of Hexokinase II on treating cells with C34, and while lactate production did decrease, this decrease was not seen to be statistically significant. As such, we believe that *E. coli* targets glycolysis through a TLR4-independent manner.

Our findings underscore the importance of the tumor microbiome in LUAD progression and metastasis. Similar interactions between bacteria and cancer cells, such as those involving *Pseudomonas aeruginosa*, have demonstrated bidirectional crosstalk with other cancer types [14]. While the relationship between bacteria and cancer remains poorly understood, studying direct interactions between tumors and their surrounding microbiome may reveal novel therapeutic targets. Incorporating synthetic microbiome models [55] in future studies could help distinguish tumor-promoting bacteria from tumor-suppressing bacteria, paving the way for innovative cancer treatments.

## Figures and Tables

**Figure 1 cancers-17-00380-f001:**
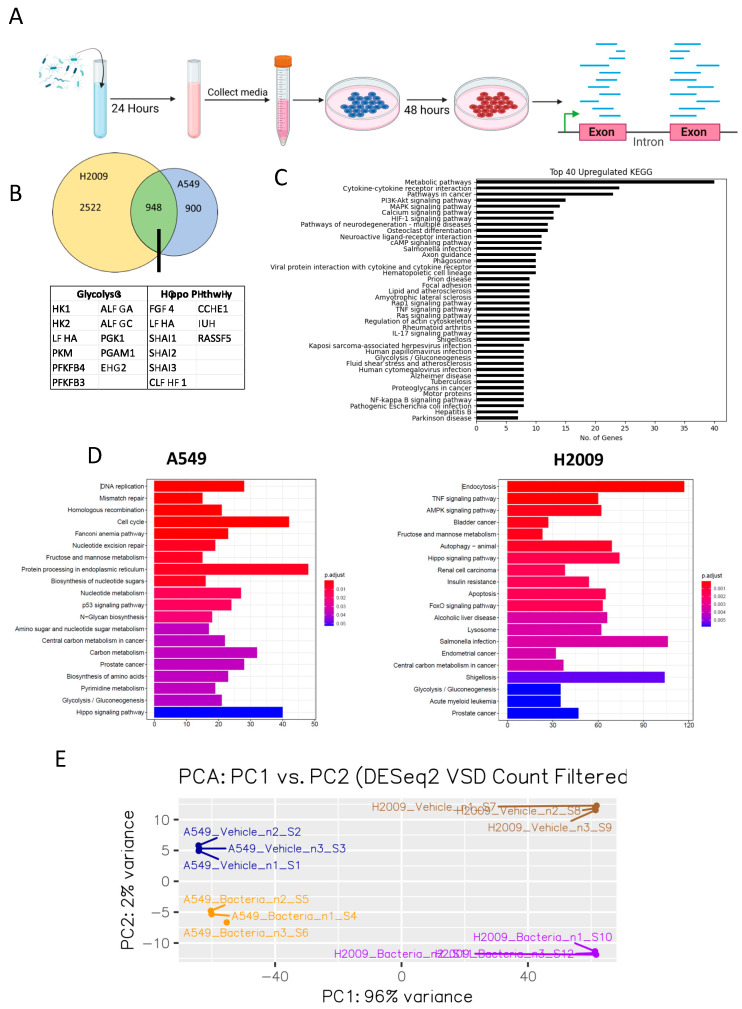
Differentially expressed genes of LUAD incubated in bacterially supplemented media. (**A**) Pipeline for analysis of RNA-seq taken from lung adenocarcinoma cell lines after bacterial biomolecule treatment. (**B**) Venn diagram representing the unique and shared upregulated LUAD genes affected under bacterial supplementation whose adjusted *p*-value < 0.05 and log2FC > 1.5. Genes related to glycolysis and the Hippo Pathway have been placed in a table below the Venn diagram. (**C**) Upregulated KEGG pathways in both LUAD cell lines. Transcriptomic data were filtered based on *p*-value < 0.05 and log2FC > 1.5. Data were then analyzed using BioKEGG (https://biopython.org/docs/1.76/api/Bio.KEGG.html, accessed on 20 January 2025) and pathways were ranked by number of transcriptomic data genes associated with a specific pathway. (**D**) Top 20 KEGG pathways enriched under the presence of *E. coli* supplementation for A549 and H2009. Adjusted *p*-values for each comparison included in the figure key. (**E**) PCA plot of RNAseq data.

**Figure 2 cancers-17-00380-f002:**
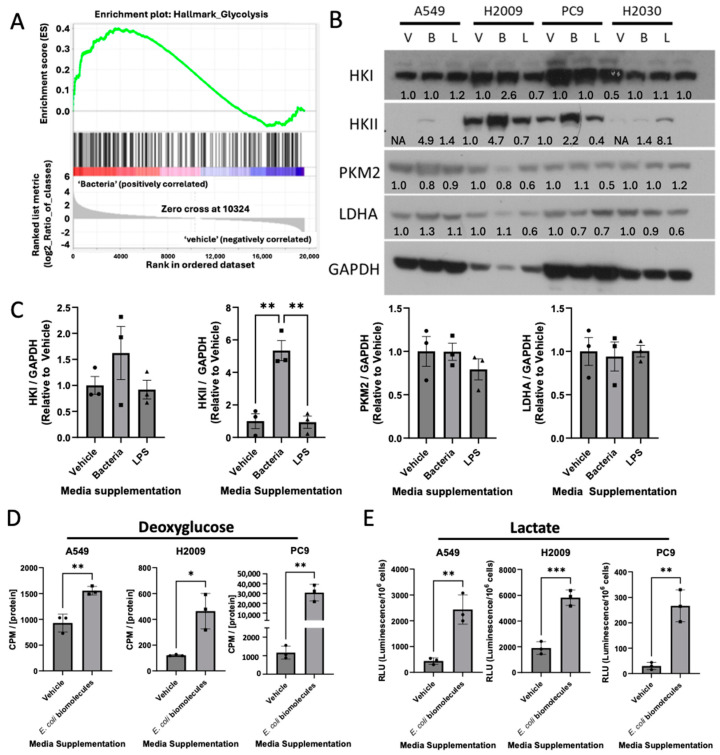
*E. coli*-supplemented media enhance glycolysis in LUAD. (**A**) Gene set enrichment analysis (GSEA) plot of representative gene sets in A549 incubated with *E. coli*-supplemented media for genes involved in glycolysis. (**B**) Western blot representing multiple glycolytic enzymes. V = vehicle (1:1, RPMI: PBS), B = bacteria (1:1, RPMI: bacterially infused media), and L = LPS (1:1, PRMI: PBS + 5 μM LPS). The uncropped bolts are shown in Appendix A. (**C**) Densitometry analysis of HKI, HKII, PKM2, and LDHA was performed using ImageJ and normalized by GAPDH. Normalized densitometry data were compared against respective LUAD cell line vehicle control. Data were analyzed using one-way ANOVA with Tukey post hoc test; n = 3 for each cell line, ** *p*-value < 0.0021. (**D**) Radiolabeled deoxyglucose readings quantified by normalizing scintillation counts per minute (CPM) to protein concentration. Data were analyzed using unpaired *t*-test; n = 3 for each cell line, * *p*-value <0.0332, ** *p*-value < 0.0021. (**E**) Lactate production of LUAD cell lines incubated in vehicle or *E. coli* biomolecules measured by luminescence and normalized by cell count. Relative luminescence units (RLUs) were analyzed using unpaired *t*-test; n = 3 for each cell lines, ** *p*-value < 0.0021, *** *p*-value < 0.0002.

**Figure 3 cancers-17-00380-f003:**
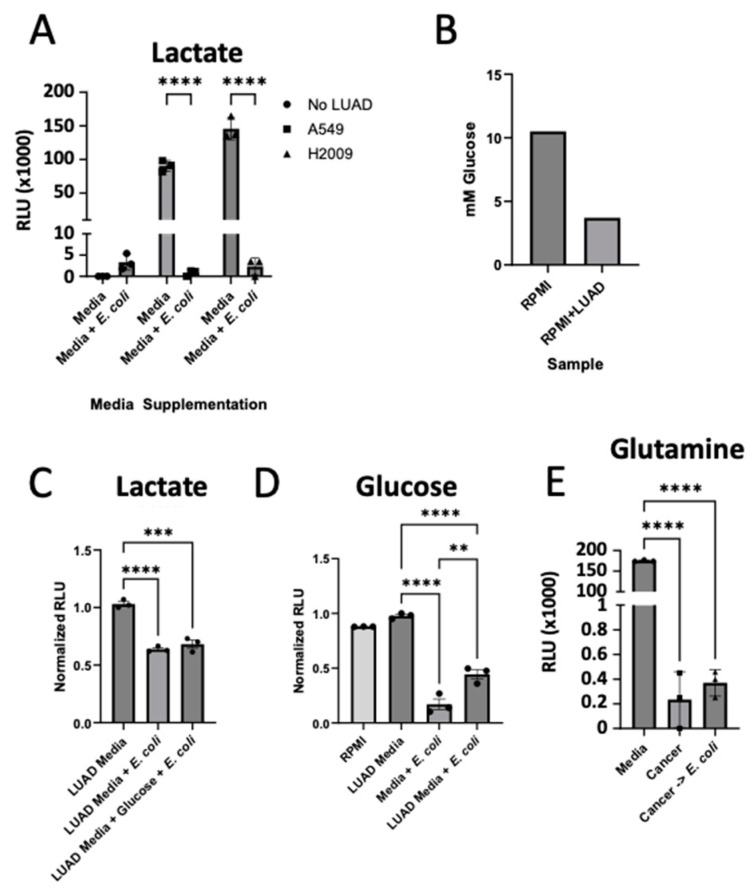
*E. coli* takes up LUAD produced lactate. (**A**) Relative luminescence units (RLUs) of lactate levels in media after 24 h of incubation without *E. coli* (media) and with *E. coli* (media + *E. coli*). Media were prepared by incubated RPMI-1640 with no LUAD (circle), A549 (square), or H2009 (triangle) for 24 h prior to addition of *E. coli*. Data were analyzed using two-way ANOVA with Sidak post hoc test (n = 3, **** *p*-value < 0.0001) with comparison of respective media with and without *E. coli*. (**B**) mM glucose levels or RPMI media and RPMI media following 24 h incubation with LUAD, as determined through glucose-glo assay. (**C**) RLU of lactate levels in media after 8 h incubation with *E. coli* with LUAD media (LUAD + *E. coli*) or LUAD media supplemented with glucose to a final concentration of 10 mM glucose (LUAD media + glucose + *E. coli*). RLUs were normalized against lactate levels in LUAD media and analyzed using one-way ANOVA with Sidak post hoc test (n = 3, *** *p*-value < 0.0002, **** *p*-value < 0.0001). (**D**) RLU of glucose levels in media after 8 h incubation with *E. coli* with RPMI-1640 media (media + *E. coli*) or LUAD media (LUAD + *E. coli*). RLUs were normalized against glucose levels in LUAD media and analyzed using one-way ANOVA with Sidak post hoc test (n = 3, ** *p*-value < 0.0021, **** *p*-value < 0.0001). (**E**) RLUs of glutamine levels in media after 24 h incubation with A549 (cancer) and incubation of LUAD media with *E. coli* for 24 h (cancer -> *E. coli*). RLUs were compared against glutamine levels found in RPMI-1640 (media) and analyzed using one-way ANOVA with Sidak post hoc test (n = 3, **** *p*-value < 0.0001).

**Figure 4 cancers-17-00380-f004:**
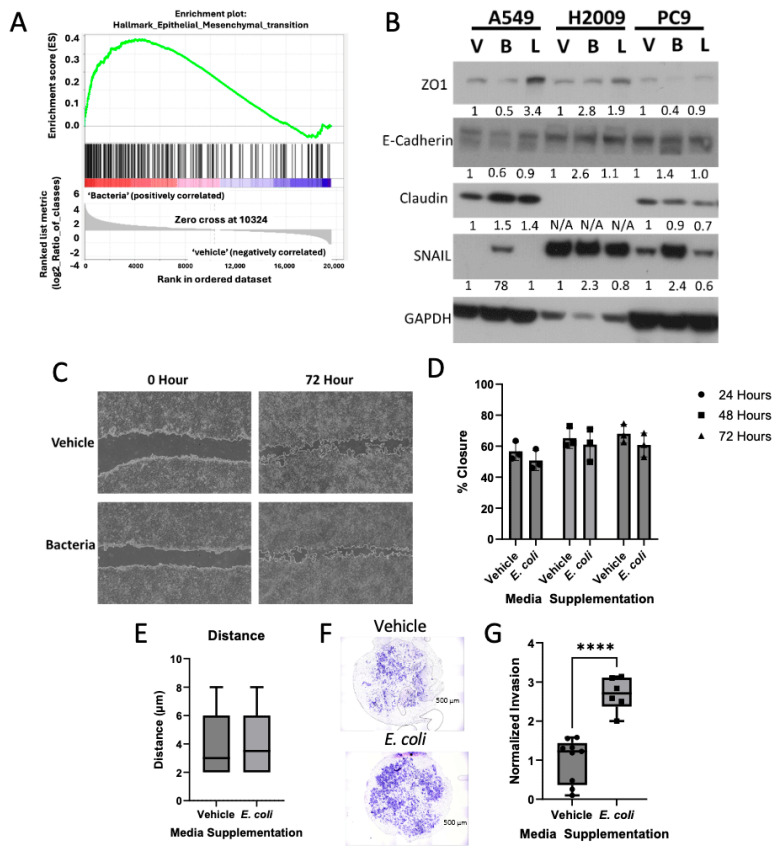
*E. coli*-supplemented media enhance invasive potential of LUAD cell lines. (**A**) Gene set enrichment analysis (GSEA) plot of representative gene sets in A549 incubated with *E. coli*-supplemented media for genes involved in epithelial-to-mesenchymal transition (EMT). (**B**) Western blot representing multiple genes involved in EMT for various LUAD cell lines incubated in vehicle (1:1, RPMI: PBS), bacteria (1:1, RPMI: bacterially infused media), and LPS (1:1, PRMI: PBS + 0.5 µg/mL LPS). Densitometry analysis was performed using ImageJ and data were normalized to GAPDH. Normalized values were compared against respective vehicle controls. The uncropped bolts are shown in Appendix A. (**C**) Representative images of wound-healing assay to determine the chemo-attractant potential of *E. coli* biomolecules. After scratch was made, Matrigel infused with *E. coli*-supplemented media was layered on top and images were taken across multiple days. (**D**) Data analysis of wound-healing assay (n = 3) of LUAD cells in either vehicle (1:1, RPMI: PBS) or media supplemented with *E. coli* (1:1, RPMI: *E. coli* media). Data were analyzed using two-way ANOVA with Sidak post hoc test. (**E**) Data of single-celled LUAD motility in either vehicle or *E. coli* after 48 h (n = 20) using Keyence Live Cell Imager. Data were analyzed using unpaired *t*-test. (**F**) Invasion assay representation of LUAD cell lines incubated with serum-starved vehicle or *E. coli*. 10% FBS serum was used as the chemo-attractant. (**G**) Data analysis of invasion assay (n = 8) of LUAD cells in either vehicle or *E. coli*. Invasion assay data were normalized by average invasion in vehicle and analyzed using Student’s *t*-test, **** *p*-value < 0.0001.

**Figure 5 cancers-17-00380-f005:**
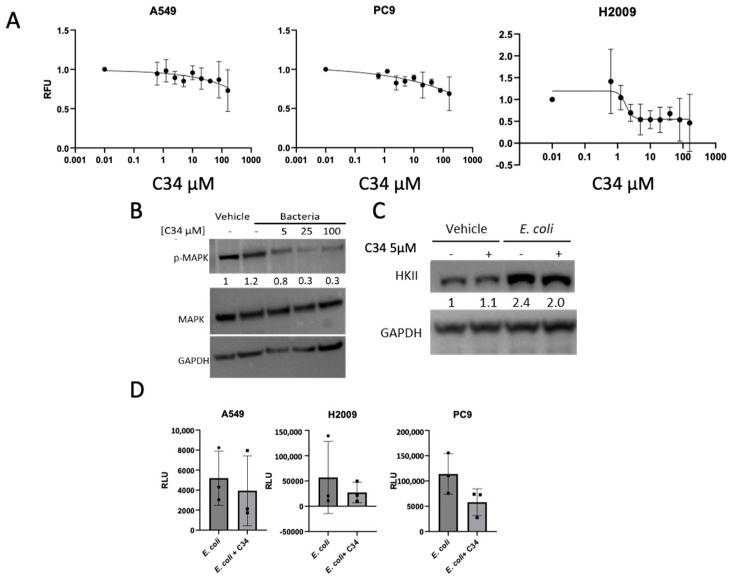
TLR4 inhibition does not impact *E. coli*-influenced glycolysis. (**A**) AlamarBlue viability assay of LUAD cell lines following treatment of increasing levels of TLR4 inhibitor, C34. LUAD cell lines were incubated for 24 h prior to incubation with alamarBlue. RFUs attained were normalized for each LUAD cell line by RFUs at lowest concentration (0.01 µM C34). (**B**) Western blot representation of A549 cell lines incubated with increasing concertation of C34. Membrane was blotted for p-MAPK and MAPK with GAPDH as a loading control along with anti-rabbit secondary. Densitometry was measured using ImageJ, normalizing by GAPDH and further normalizing p-MAPK by respective MAPK. Normalized values were compared to p-MAPK densitometry values of vehicle without C34. The uncropped bolts are shown in Appendix A. (**C**) Western blot representation of A549 cell lines incubated with *E. coli* biomolecules (*E. coli*)/without (vehicle) and with/without 5 µM C34. Membrane was blotted for HKII, with GAPDH used as a loading control. Densitometry was analyzed using ImageJ, with values normalized to GAPDH and normalized values relative to vehicle without 5 µM C34. (**D**) Relative luminescent units (RLUs) of lactate measurement in media of A549, H2009, and PC9 incubated with *E. coli* biomolecules with and without 5 µM C34. Data were analyzed using unpaired *t*-test, n = 3.

**Figure 6 cancers-17-00380-f006:**
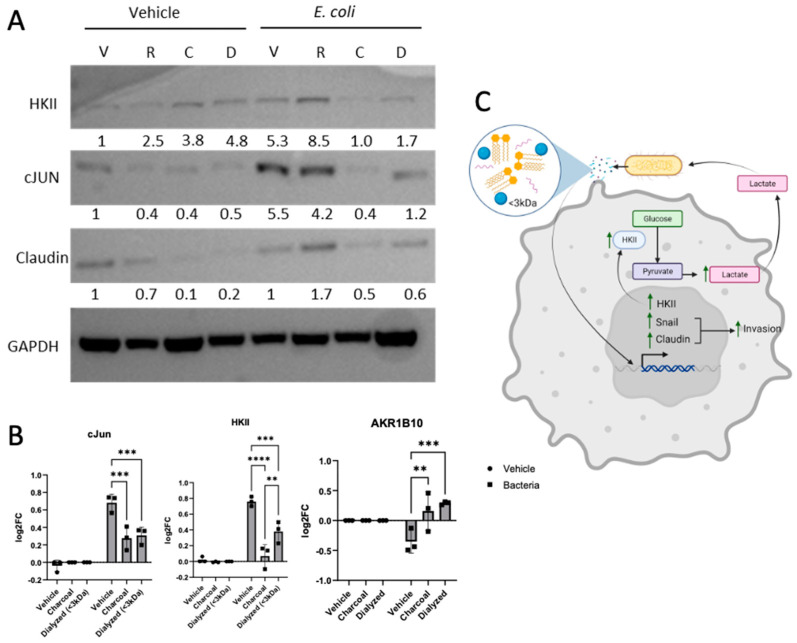
Treatment of *E. coli*-supplemented media further impacts LUAD. (**A**) Western blot representation of A549 incubated in vehicle or *E. coli*-supplemented media untreated (V) or treated with RNAse (R) or dextran-coated charcoal (C) or dialyzed to remove molecules <3 kDa (D). Membrane was blotted for HKII, CJUN, and Claudin, with GAPDH used as a loading control. Densitometry was analyzed using ImageJ, wherein densitometry values were normalized to respective GAPDH and normalized values were compared against A549 incubated in untreated vehicle media. The uncropped bolts are shown in Appendix A. (**B**) RT-qPCR analysis of A549 cells incubated in vehicle or *E. coli*-supplemented media (bacteria) untreated (vehicle) or treated with dextran-coated charcoal or dialyzed to remove molecules <3 kDa. Data were analyzed using a two-way ANOVA with a Tukey post hoc test (n = 3, ** *p*-value < 0.0021, *** *p*-value < 0.0002, and **** *p*-value < 0.0001). (**C**) Graphical representation of *E. coli* secreting biomolecules which LUAD can take up, resulting in altered phenotypes.

## Data Availability

RNAseq datasets have been GEO; Accession GSE286573.

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
