# Peer review of "E. coli Biomolecules Increase Glycolysis and Invasive Potential in Lung Adenocarcinoma"

_cancers, 2025, doi:10.3390/cancers17030380_

Round 1
Reviewer 1 Report
Comments and Suggestions for Authors
The authors of the manuscript “E. coli biomolecules increases glycolysis and invasive potential in lung adenocarcinoma” investigates the molecular mechanism of the pathologic consequences of the presence of E. coli in the microbiome of lung adenocarcinoma. The data presented herein are derived from a cell culture model. The use of three cell lines ensures that reliable results are obtained. The authors present data supporting their final conclusion that E.coli bacterial factors mainly affect glycolysis in lung cancer cells. Although the experimental data are convincing and worthy of publication, the authors should spend time on more careful phrasing of thoughts, use of terms, and generally correcting English grammar.
There are a few places in the text where the singular and plural forms of verbs are used incorrectly. Even in the title! “E. coli biomolecules increase glycolysis and invasive potential in lung adenocarcinoma”.
It is more correct to use the term “bacterial factors” instead of “biomolecules” since E.coli can only have biological, not synthetic molecules!
Line 298. It is more correct to use the word “upregulated” instead of “… positively enriched..”, as there could be no “negatively enriched” something!
In the Introduction(lines 47-50) it is said “While previously believed to consist of mammalian cells that were reprogrammed by the tumor, it has come to light that other organisms also influence the TME, namely fungi and bacteria”. It sounds as if the tumor does not appear to be made up of mammalian cells and is some kind of an alien organism! It would be better to rephrase! For example, like this: “While previously believed to consist of mammalian cells that were reprogrammed during tumorigenesis, it emerged that other organisms constituting the TME, specifically fungi and bacteria, also influence this process”.
Line 59-61. (“To add to the complexity of the microbiome, previous studies have shown that the presence tumors modulate the surrounding microbiome (Jin et al., 2019;Vega et al., 2023)”). While it is argued here that it is tumor that affects microbiota, the references-affirm the opposite, that microbiota affects tumor and this is more in line with the interest of the readers. It should be rephrased.
Lines 88,89, it is unclear whether you are talking about RNA species or indeed the enzyme RNAse! “We observed removal of RNAse further enhanced cJUN protein expression while charcoal and dialysis both resulted in decreased cJUN protein”.
Maybe, it was meant to say: “We observed removal of RNA species by Range further enhanced cJUN protein expression while charcoal and dialysis both resulted in decreased cJUN protein expression”?!
Line 203. It is more accurate to write “Cell lysis was performed using…”
Line 253. The meaning of the sentence on lines 253-255 will be clearer if you write it as follows ”We do not consider that the influence of E. coli bacterial factors on some of these pathways is an artifact of the selection we performed, as some of these pathways are also among the top 20 most upregulated KEGG pathways of individual LUAD cell lines (Supplementary Figure 1).
Line 200. “…using the BZX-800 analyser program”.
Line 273. “…by isolating genes upregulated …..”
Also, using cell names/any abbreviations without indicating that they are cells/other subjects is a sign of sloppiness. Please, correct it in the main text and in the supplementary material. For example, in lines 228-229, it should be:
DEGs observed in A549 and H2009 cells were combined, sorted, and ranked using Python (version 3.9.13), with pandas (version 1.4.2), matplotlib (version 3.5.1), and bioservices(version 1.11.2).
Or, in lines 443, it should be “By performing an invasion assay using transwell chambers, wherein FBS was used as our chemoattractant and our LUAD cells were ………
In lines 292,293 (With this in mind, we chose to investigate the effects of these E.coli biomolecules on EMT”). Which “these” if the text above it was about the cellular EMT factors?!
In lines 299-301 (“As our RNA-seq data showed elevated transcriptional levels of several glycolytic enzymes, we next investigated whether these genes were also upregulated translationally”), information would be more informative and meaningful if rephrased, for example, like this: As our RNA-seq data showed elevation of several glycolytic enzymes at the gene expression level, we next investigated whether these genes were also upregulated at protein level.
In line 301-305(“As LPS is a major constituent of gram negative bacterial cell wall, and LPS has previously been shown to increase glycolysis in colorectal cancer (Wu et al.,2021b), we chose to incorporate a group containing 0.5μg/ml of LPS to determine whether LPS was the driving force of increased glycolytic expression in our LUAD cells”). A group of what? A group of samples? And further, maybe, “…instead of increased glycolytic expression…” it is better to use”… increased expression of glycolytic genes”?!
Line 336-337. Interestingly, we observed a dramatic increase in lactate in all three……”
Line 359. “As our RNA-seq data reveal upregulation of genes controlling several reversible steps in glycolysis….”
Line 364. “While our data revealed depletion of glutamine in the media from LUAD cell lines, we did not observe glutamine levels are affected by subsequent culturing of LUAD cells in the bacteria supplemented RPMI-1640 media (Figure 3E).”
Line 438. “… to breast cancer transplants…”.
Line 441.”… and live cell imaging data…”
Line 465-467. It would be more correct to rephrase the sentence in these lines as follows: “Previous investigators have reported a reduction in the progression of palmitic acid-induced colorectal cancer after administration of the C34 chemical at a concentration of 5 μM.”
Line 588. Pseudomonas aeruginosa.
Please use “specifically” instead of “namely”, “observed” instead of “witnessed”, “microbiota” instead of “microbiome” everywhere in the text if you are describing the action of microorganisms rather than a set of nucleotide sequences of microorganisms, “the Hippo pathway” instead of “the hippo” or “The Hippo Pathway”, “based on these observations” instead of ”As such”.
In general, I would recommend the help of a native English speaker in proofreading the text.
It would have been nice if the authors had stuck to the same terms throughout the text, in the Material and Method section, Results and Figure legends. For example, In the chapter 2.1.Cell Culture, it would be good to continue the sentence “Following incubation, E.coli cells were removed from the media by centrifugation (12,000xg for 10 minutes) and the supernate was sterilized (0,2 μ). This media is called “Bacteria supplemented RPMI-1640 media”.
The designation of the chemicals used in the study must be correct. Line 169 should clearly indicate that [3H]-labeled deoxyglucose was used.
“[3H]-labeled deoxyglucose (PerkinElmer, Waltham, MA, USA) was added to LUAD cell lines at 2 μCi/ml following incubation for one hour”.
Line 176. Instead of “LUAD cell lines were incubated in 6-well dish at a confluence of 2x106 overnight”, it would be more accurate to write “LUAD cell lines were seeded in a 6-well dish at a density of 2x106/well the night before the experiment.”
Lines 86,195: 1x105 and 5x104 are absolute numbers, not a density designation.
It would be more accurate to write “For investigating chemoattractant capacity of E.coli bacterial factors, LUAD cells were seeded on the insert at a density of 1x105/(area of insert in)cm2”.
Line 192. I do not think something could be hydrolysed in 1xPBS! Either one can use the term “hydrated” or “dissolved”. In any case, it would be a good idea to limit Chapter 2.7 to lines 183-192 and call it just “Invasion Assay”. It is not necessary to include the experiment described in lines 192-200. First of all, it is overly sophisticated and uninformative (It was negative and data were not presented). In addition, there will be no need then to devote too much text to it in the Discussion (cumbersome to understand). The message in lines 557-567 could be phrased, for example, as follows: “We next investigated whether bacterial factors secreted by E.coli could act as chemoattractants for LUAD cells. Our invasion assay showed that the invasive potential of LUAD cells increased dramatically towards to a known chemoattractant, fetal bovine serum (FBS), when LUAD cells were serum starved and E.coli factors were added. Our results support previously published data that it is the local cancer microbiota that stimulates metastasis, not the microbiota in the gut.”
Note! A microbiome is a collection of microbial genes. A microbiota is a collection of microorganisms.
Figure 1. Where is 1D panel, as it is mentioned in the figure legend?
Figure 2D, on the Y axis to add “Deoxyglucose”. 2E-on the Y axis to add “Lactate”.
On the X axis- “E.coli factors” instead of “E.coli biomolecules”.
There are two variants of the legend of figure 2E. There is no data confirming the second variant of the legend, although there is a mention of this variant in the text (lines 336-337).
Figure 3A, 3C on the Y axis to add “Lactate”, Figure 3D on the Y axis to add “Glucose”, Figure 3E to add “Glutamine”.
Texts from the Result section, lines 339-342, 368-371, 436-438, is more appropriate to move to the Discussion.
In addition, it would be good to speculate in the Discussion about the discrepancies in LDHA expression levels in LUAD cells when exposed to E.coli observed by different assays. It is upregulated accordingly to RNAseq data(Figure 1), unchanged in immunoblotting analysis(Figure 2C) and, apparently, increased in luminescence assay(Figure 3A).
No rationale is given for the focus specifically on E. coli in the pathogenesis of LUAD, except for mentioning the role of Gram-negative bacteria. The authors could at least refer to the work of Wong et al.
Wong LM, Shende N, Li WT, Castaneda G, Apostol L, Chang EY, Ongkeko WM. Comparative Analysis of Age- and Gender-Associated Microbiome in Lung Adenocarcinoma and Lung Squamous Cell Carcinoma. Cancers (Basel). 2020 Jun 2;12(6):1447. doi: 10.3390/cancers12061447. PMID: 32498338; PMCID: PMC7352186.
Otherwise, in the future, it would be good to study the molecular mechanism caused by the bacteria reported to be most commonly found in LUAD tissues.
By the way, specify E.coli strain which was used in the study?
Comments on the Quality of English LanguageThe authors present data supporting their final conclusion that E.coli bacterial factors mainly affect glycolysis in lung cancer cells. Although the experimental data are convincing and worthy of publication, the authors should spend time on more careful phrasing of thoughts, use of terms, and generally correcting English grammar.
Author Response
We would like to express our sincere appreciation to the reviewer for taking the time to thoroughly edit our manuscript and provide detailed suggestions. You will see that we have taken much time to address these comments with much care. We have fixed nearly all issues raised by the reviewer and we have completely edited the manuscript to fix the english and grammar. The work is much improved due to the diligence of the reviewer.
Reviewer comments:
There are a few places in the text where the singular and plural forms of verbs are used incorrectly. Even in the title! “E. coli biomolecules increase glycolysis and invasive potential in lung adenocarcinoma”. fixed
It is more correct to use the term “bacterial factors” instead of “biomolecules” since E.coli can only have biological, not synthetic molecules! We have chosen to stick with the term "biomolecules" since the definition is "organic molecules produced by living organisms", which we feel exactly encompasses what we are studying.
Line 298. It is more correct to use the word “upregulated” instead of “… positively enriched..”, as there could be no “negatively enriched” something! fixed
In the Introduction(lines 47-50) it is said “While previously believed to consist of mammalian cells that were reprogrammed by the tumor, it has come to light that other organisms also influence the TME, namely fungi and bacteria”. It sounds as if the tumor does not appear to be made up of mammalian cells and is some kind of an alien organism! It would be better to rephrase! For example, like this: “While previously believed to consist of mammalian cells that were reprogrammed during tumorigenesis, it emerged that other organisms constituting the TME, specifically fungi and bacteria, also influence this process”. fixed
Line 59-61. (“To add to the complexity of the microbiome, previous studies have shown that the presence tumors modulate the surrounding microbiome (Jin et al., 2019;Vega et al., 2023)”). While it is argued here that it is tumor that affects microbiota, the references-affirm the opposite, that microbiota affects tumor and this is more in line with the interest of the readers. It should be rephrased. fixed
Lines 88,89, it is unclear whether you are talking about RNA species or indeed the enzyme RNAse! “We observed removal of RNAse further enhanced cJUN protein expression while charcoal and dialysis both resulted in decreased cJUN protein”. added the word "enzyme", as we are talking about RNAse. also fixed the wording.
Maybe, it was meant to say: “We observed removal of RNA species by Range further enhanced cJUN protein expression while charcoal and dialysis both resulted in decreased cJUN protein expression”?! fixed
Line 203. It is more accurate to write “Cell lysis was performed using…” fixed
Line 253. The meaning of the sentence on lines 253-255 will be clearer if you write it as follows ”We do not consider that the influence of E. coli bacterial factors on some of these pathways is an artifact of the selection we performed, as some of these pathways are also among the top 20 most upregulated KEGG pathways of individual LUAD cell lines (Supplementary Figure 1). fixed
Line 200. “…using the BZX-800 analyser program”. fixed
Line 273. “…by isolating genes upregulated …..” fixed
Also, using cell names/any abbreviations without indicating that they are cells/other subjects is a sign of sloppiness. Please, correct it in the main text and in the supplementary material. For example, in lines 228-229, it should be:
DEGs observed in A549 and H2009 cells were combined, sorted, and ranked using Python (version 3.9.13), with pandas (version 1.4.2), matplotlib (version 3.5.1), and bioservices(version 1.11.2). fixed
Or, in lines 443, it should be “By performing an invasion assay using transwell chambers, wherein FBS was used as our chemoattractant and our LUAD cells were ……… fixed
In lines 292,293 (With this in mind, we chose to investigate the effects of these E.coli biomolecules on EMT”). Which “these” if the text above it was about the cellular EMT factors?! fixed
In lines 299-301 (“As our RNA-seq data showed elevated transcriptional levels of several glycolytic enzymes, we next investigated whether these genes were also upregulated translationally”), information would be more informative and meaningful if rephrased, for example, like this: As our RNA-seq data showed elevation of several glycolytic enzymes at the gene expression level, we next investigated whether these genes were also upregulated at protein level. fixed
In line 301-305(“As LPS is a major constituent of gram negative bacterial cell wall, and LPS has previously been shown to increase glycolysis in colorectal cancer (Wu et al.,2021b), we chose to incorporate a group containing 0.5μg/ml of LPS to determine whether LPS was the driving force of increased glycolytic expression in our LUAD cells”). A group of what? A group of samples? And further, maybe, “…instead of increased glycolytic expression…” it is better to use”… increased expression of glycolytic genes”?! fixed wording
Line 336-337. Interestingly, we observed a dramatic increase in lactate in all three……” fixed
Line 359. “As our RNA-seq data reveal upregulation of genes controlling several reversible steps in glycolysis….” fixed
Line 364. “While our data revealed depletion of glutamine in the media from LUAD cell lines, we did not observe glutamine levels are affected by subsequent culturing of LUAD cells in the bacteria supplemented RPMI-1640 media (Figure 3E).” fixed
Line 438. “… to breast cancer transplants…”. fixed
Line 441.”… and live cell imaging data…” fixed
Line 465-467. It would be more correct to rephrase the sentence in these lines as follows: “Previous investigators have reported a reduction in the progression of palmitic acid-induced colorectal cancer after administration of the C34 chemical at a concentration of 5 μM.” fixed
Line 588. Pseudomonas aeruginosa. fixed
Please use “specifically” instead of “namely”, “observed” instead of “witnessed”, “microbiota” instead of “microbiome” everywhere in the text if you are describing the action of microorganisms rather than a set of nucleotide sequences of microorganisms, “the Hippo pathway” instead of “the hippo” or “The Hippo Pathway”, “based on these observations” instead of ”As such”. fixed
In general, I would recommend the help of a native English speaker in proofreading the text. done
It would have been nice if the authors had stuck to the same terms throughout the text, in the Material and Method section, Results and Figure legends. For example, In the chapter 2.1.Cell Culture, it would be good to continue the sentence “Following incubation, E.coli cells were removed from the media by centrifugation (12,000xg for 10 minutes) and the supernate was sterilized (0,2 μ). This media is called “Bacteria supplemented RPMI-1640 media”. fixed
The designation of the chemicals used in the study must be correct. Line 169 should clearly indicate that [3H]-labeled deoxyglucose was used. fixed
“[3H]-labeled deoxyglucose (PerkinElmer, Waltham, MA, USA) was added to LUAD cell lines at 2 μCi/ml following incubation for one hour”. fixed
Line 176. Instead of “LUAD cell lines were incubated in 6-well dish at a confluence of 2x106 overnight”, it would be more accurate to write “LUAD cell lines were seeded in a 6-well dish at a density of 2x106/well the night before the experiment.” fixed
Lines 86,195: 1x105 and 5x104 are absolute numbers, not a density designation. fixed
It would be more accurate to write “For investigating chemoattractant capacity of E.coli bacterial factors, LUAD cells were seeded on the insert at a density of 1x105/(area of insert in)cm2”. fixed
Line 192. I do not think something could be hydrolysed in 1xPBS! Either one can use the term “hydrated” or “dissolved”. In any case, it would be a good idea to limit Chapter 2.7 to lines 183-192 and call it just “Invasion Assay”. It is not necessary to include the experiment described in lines 192-200. First of all, it is overly sophisticated and uninformative (It was negative and data were not presented). In addition, there will be no need then to devote too much text to it in the Discussion (cumbersome to understand). The message in lines 557-567 could be phrased, for example, as follows: “We next investigated whether bacterial factors secreted by E.coli could act as chemoattractants for LUAD cells. Our invasion assay showed that the invasive potential of LUAD cells increased dramatically towards to a known chemoattractant, fetal bovine serum (FBS), when LUAD cells were serum starved and E.coli factors were added. Our results support previously published data that it is the local cancer microbiota that stimulates metastasis, not the microbiota in the gut.” fixed
Note! A microbiome is a collection of microbial genes. A microbiota is a collection of microorganisms. changed all to "microbiome" since it is defined as "the microbiota, genes, metabolites and other factors produced by the microbiota"
Figure 1. Where is 1D panel, as it is mentioned in the figure legend? fixed
Figure 2D, on the Y axis to add “Deoxyglucose”. 2E-on the Y axis to add “Lactate”. added
On the X axis- “E.coli factors” instead of “E.coli biomolecules”. did not change
There are two variants of the legend of figure 2E. There is no data confirming the second variant of the legend, although there is a mention of this variant in the text (lines 336-337). deleted incorrect one
Figure 3A, 3C on the Y axis to add “Lactate”, Figure 3D on the Y axis to add “Glucose”, Figure 3E to add “Glutamine”. added
Texts from the Result section, lines 339-342, 368-371, 436-438, is more appropriate to move to the Discussion. left as is
In addition, it would be good to speculate in the Discussion about the discrepancies in LDHA expression levels in LUAD cells when exposed to E.coli observed by different assays. It is upregulated accordingly to RNAseq data(Figure 1), unchanged in immunoblotting analysis(Figure 2C) and, apparently, increased in luminescence assay(Figure 3A). did not address
No rationale is given for the focus specifically on E. coli in the pathogenesis of LUAD, except for mentioning the role of Gram-negative bacteria. The authors could at least refer to the work of Wong et al. We focused on e. coli based on our previous work.
Reviewer 2 Report
Comments and Suggestions for Authors
Major:
1. The effect of RNAse treatment on cJUN expression is unclear. The authors claimed that RNAse treatment leads to an increase in cJUN expression, but the study does not rule out the possibility that RNAse itself directly affects cancer cells, independent of bacterial biomolecules. It's better to include control experiments, such as treating the cells with RNAse alone (without bacterial biomolecules), to confirm whether the upregulation of cJUN is caused by the RNAse treatment itself or by its degradation of RNA from bacterial biomolecules.
2. Another issue is the insufficient analysis of bacterial biomolecules. Although the authors used methods like charcoal treatment and dialysis to identify potential active components, the precise nature of the key biomolecules responsible for the observed effects remains undefined. It is suggested to include chemical characterization (e.g., mass spectrometry analysis) of the bacterial biomolecule mixture to identify specific active compounds contributing to glycolysis and invasive potential in LUAD cells.
3. The authors should better visualize their RNA-seq experiments in a PCA plot so that the treatment can be better conveyed.
4. The authors should deposit their RNA-seq data in public databases like GEO and provide public links.
5. It seems the authors misplaced the subfigures labels in Figure 1 – the legend has the (D) panel but the figure doesn’t. The authors should double check the figure layout and legends to make sure they are consistent.
Minor:
6. “Transcriptomic data was combined and filtered based on p-value<0.05” in Line 278-279 is vague. If the authors are referring to the enrichment analysis, it shouldn’t be the “transcriptomic data”, but “enriched KEGG terms” are “combined and filtered”. And the authors should clarify if they are using adjusted p-values or the raw ones.
7. Gene symbols should be italicized in this manuscript.
Author Response
We are extremely grateful for the comments and suggestions provided by the reviewer. We have addressed the comments and the manuscript is greatly improved because of the comments.
- The effect of RNAse treatment on cJUN expression is unclear. The authors claimed that RNAse treatment leads to an increase in cJUN expression, but the study does not rule out the possibility that RNAse itself directly affects cancer cells, independent of bacterial biomolecules. It's better to include control experiments, such as treating the cells with RNAse alone (without bacterial biomolecules), to confirm whether the upregulation of cJUN is caused by the RNAse treatment itself or by its degradation of RNA from bacterial biomolecules. -This point was clarified in the text. RNAase treatment does not alter JUN expression. bacteria media increases JUN levels and RNAse does not block this increase, whereas charcoal and dialysis removes a factor from the bacteria media that causes increase in JUN. All controls are in the figure.
- Another issue is the insufficient analysis of bacterial biomolecules. Although the authors used methods like charcoal treatment and dialysis to identify potential active components, the precise nature of the key biomolecules responsible for the observed effects remains undefined. It is suggested to include chemical characterization (e.g., mass spectrometry analysis) of the bacterial biomolecule mixture to identify specific active compounds contributing to glycolysis and invasive potential in LUAD cells. -This will be the focus of much work in the future and is beyond the scope of this publication.
- The authors should better visualize their RNA-seq experiments in a PCA plot so that the treatment can be better conveyed. Added
- The authors should deposit their RNA-seq data in public databases like GEO and provide public links. -Our bioinformatics team has submitted the paperwork to ensure this happens.
- It seems the authors misplaced the subfigures labels in Figure 1 – the legend has the (D) panel but the figure doesn’t. The authors should double check the figure layout and legends to make sure they are consistent. -Legends have been corrected.
Minor:
- “Transcriptomic data was combined and filtered based on p-value<0.05” in Line 278-279 is vague. If the authors are referring to the enrichment analysis, it shouldn’t be the “transcriptomic data”, but “enriched KEGG terms” are “combined and filtered”. And the authors should clarify if they are using adjusted p-values or the raw ones. all analyses are based off of adjusted p-values. we have adjusted the text.
- Gene symbols should be italicized in this manuscript. -Fixed
Round 2
Reviewer 1 Report
Comments and Suggestions for Authors
The answer to the question about the specificity of the E.coli strain is not provided.
There are still small faults throughout the text.
The authors left “the hippo” instead of “the Hippo pathway”, although a correction was suggested and although they reply here that it has been corrected(line 525).
In general, it is advisable to consult a native English speaker in the future to avoid “witnessed” instead of “observed”, “modulated” instead of “modified”, etc.
Changes in glycolysis are not so much seen as phenotypic changes as metabolic changes(line 556).